# Plant Growth Regulators: An Overview of WOX Gene Family

**DOI:** 10.3390/plants13213108

**Published:** 2024-11-04

**Authors:** Haroon Rasheed, Lin Shi, Chichi Winarsih, Bello Hassan Jakada, Rusong Chai, Haijiao Huang

**Affiliations:** 1State Key Laboratory of Tree Genetics and Breeding, Northeast Forestry University, Harbin 150040, China; haroongb@nefu.edu.cn (H.R.); sl1256544977@163.com (L.S.); cwinarsih@nefu.edu.cn (C.W.); 2Key Laboratory of Saline-Alkali Vegetation Ecology Restoration, Ministry of Education, College of Life Sciences, Northeast Forestry University, Harbin 150040, China; bello.jakada@nefu.edu.cn; 3Forest Botanical Garden of Heilongjiang Province, Haping Road 105, Harbin 150040, China

**Keywords:** WOX genes, abiotic stress, plant growth and development, transcription factor, biotechnology

## Abstract

The adaptation of plants to land requires sophisticated biological processes and signaling. Transcription factors (TFs) regulate several cellular and metabolic activities, as well as signaling pathways in plants during stress and growth and development. The WUSCHEL-RELATED HOMEOBOX (WOX) genes are TFs that are part of the homeodomain (HD) family, which is important for the maintenance of apical meristem, stem cell niche, and other cellular processes. The WOX gene family is divided into three clades: ancient, intermediate, and modern (WUS) based on historical evolution linkage. The number of WOX genes in the plant body increases as plants grow more complex and varies in different species. Numerous research studies have discovered that the WOX gene family play a role in the whole plant’s growth and development, such as in the stem, embryo, root, flower, and leaf. This review comprehensively analyzes roles of the WOX gene family across various plant species, highlighting the evolutionary significance and potential biotechnological applications in stress resistance and crop improvement.

## 1. Introduction

As sessile organisms, plants are extremely susceptible to various abiotic stresses such as cold, heat, drought, and salinity [1]. Abiotic stresses cause 50 to 70% yield losses in horticultural crops [2]. According to a 2015 Food and Agricultural Organization report, 50% of commercial agriculture will be affected due to global warming and climate change by the end of this century. Also, a total of 20–30% of the agricultural soils are affected by salinity globally [3]. Plants have evolved an extensive network of regulatory systems, which involve reprogramming the transcriptional and post-transcriptional expression of many genes to cope with environmental stress. Transcription factors (TFs) are crucial for the transcriptional and post-transcriptional control of genes involved in response to environment stress response [1].

The WOX TFs play a significant role in plant growth and development and biotic and abiotic stresses responses. For example, it was found that *WOX2* increases the expression of the auxin transporter *PIN1 gene* to regulate stem cell organogenesis in *Arabidopsis* [4]. *PpWOX2* (*Pinus pinaster*) overexpression enhanced somatic embryogenesis and plant organ formation in *Arabidopsis* transgenic seedlings [5]. By evaluating the phenotypes and gene expression patterns of *Hordeum vulgare WOX3* and *NARROW LEAFED DWARF1* (*NLD1*), we could speculate that both *HvWOX3* and *HvNLD1* promote leaf bilateral outgrowth and trichome formation in barley [6]. Also, *WOX4* and *BRI1*-*EMS*-*SUPPRESSOR1* (*BES1*) are involved in various internal signals and responses to external stimuli. *WOX11* directly targets the auxin signaling pathway, is expressed in the founder cell, and controls its establishment during de novo root regrowth and callus formation [7]. *WOX5* protein moves from the root niche organizer, the quiescent center, into the columella stem cells in the *Arabidopsis* root meristem, where it directly represses the transcription factor gene *CDF4* [8]. The overexpression of *WOX14* led to an increase in the generation of bioactive gibberellins and caused radial development in vascular stem tissues with significant lignification [9]. Research has been conducted to analyze loss-of-function genes by using the CRISPR/Cas9 genome editing tool to investigate the role of WOX genes in plant development. Loss of function of the *SILAM1* (*WOX1*) gene in tomato led to narrow leaves and secondary leaflet reduction [10]. In walnut, the knockout of the *JrWOX11* gene using the CRISPR/Cas9 genome editing tool lead to significant reduction in adventitious root formation [11]. To date, several studies have also reported the effectiveness of WOX genes in adapting to various abiotic stresses. In *Arabidopsis*, freezing tolerance is favorably controlled by *AtHOS9* (*AtWOX6*) **[12]**. In rice, overexpression of *OsWOX11* can regulate root growth and increase its ability to withstand potassium scarcity [13]. Similarly, *OsWOX13* overexpression in rice, triggered by the *rab21* promoter, improves drought tolerance and shortens the blooming period by 7–10 days [14]. It has been demonstrated that overexpressing *WOX5* of *Jatropha curcas* in rice increases drought stress tolerance [15]. Poplar salt tolerance can be regulated by *PagWOX11/12a* by inducing the expression of the *PagCYP736A12* gene [16]. Transgenic poplar with overexpressed *JrWOX11* enhanced resistance to salt and osmotic stress [17]. *GhWOX4* controls the vascular system development of cotton, improving the resistance to drought stress [18]. While *WOX2* is important in regulating plant organogenesis and somatic embryogenesis in *Arabidopsis*, a study in mulberry shows *WOX2* is highly expressed in apical buds and lowly expressed both in leaf and stem [19].

The WOX gene family structure and development activities have been widely investigated in model plants like *Arabidopsis*, but more studies are needed on woody plants [20]. Limited genomic data in non-model plants typically make it difficult to identify *WOX* genes. Furthermore, investigating WOX gene regulation in crop plants like rice and maize reveals considerable changes in gene expression patterns compared to model species like *Arabidopsis thaliana.* Much research focuses on individual WOX genes, but the entire spectrum of *WOX* gene interaction networks and their functions in many signaling cascades remains unexplored. WOX genes interact with various other regulatory proteins, including CLAVATA, and their partners vary between plant species, resulting in different developmental consequences [21]. The role of WOX genes in plant responses to environmental challenges such as drought, salt, and heavy metals is currently under investigation. However, more research is needed to completely understand how WOX genes affect abiotic stress responses, and understanding the WOX family is crucial for developing the next generation of plant crops.

## 2. Classification and Identification of *WOXs*

*WOX* gene family was first discovered in *Arabidopsis thaliana* in 1996, with its essential role in shoot and floral development [22]. The term “HOMOBOX” refers to a gene that can cause a single component of the cell to undergo metamorphosis when it develops from one embryo stage to another which was first proposed by William Bateson [23]. WOXs are part of the homeodomain (HD) family [24]. The *WOX* transcription factors were classified into 14 subfamilies, which include *WOX* (homeobox related to WUSCHEL), BELL (homeodomain similar to BELL), PINTOX, NDX (homeobox NODULIN), KNOX (homeobox KNOTTED like), LD (luminidependens homeodomain), PHD (plant homeodomain with a finger domain), HD-ZIP I–IV (homeodomain leucine zipper), ZF-HD (zinc finger homeodomain), and DDT (homeodomain DDT) [25]. High-affinity monomers of HDs attach to DNA by interactions with the helix–turn–helix (HTH) structure [1]. Plants maintain the HD in the *WOX* N-terminal [26]. As shown in Figure 1, according to evolutionary history, WUSCHEL-related homeobox is divided into the modern/WUS clade, intermediate clade, and ancient clade. Phylogenetic study indicates that the WUS clade originated in the common ancestor of spermatophytes and Polypodiidae, while members of the intermediate clade come into view in lycopodiophytes and in seed plants [27,28,29]. The *WOX* genes of the ancient clades originate in Embryophyta and chlorophyte genomes [30]. Three peptide motifs in the homeodomain (HD) were identified as markers of different super clades; NVFYWEQNH for the modern clade, NVFYWFQNR for the intermediate clade, and NVYNWFQNR for the ancient clade [28,31]. Among 15 *WOX* genes in *Arabidopsis*, the protein of the modern clade is the largest which contains *WUS*, *WOX1*/6, *WOX2*, *WOX3*, *WOX4*, and *WOX5*/*7* subclades and, in the modern/WUS clade, *WOX* genes also contain conserved domains such as the ERF-associated amphiphilic repression (EAR) domain and WUS motifs other than the homeodomain [32]. The protein of the intermediate clade contains *WOX8/9* and *WOX11/12* subclades and the ancient clade contains *WOX 10/13/14* [33]. The ancient clade *WOX* genes are expressed differently in different species, which shows the function of ancient clade *WOX* genes is species-specific [34]. The *AtWOX14* gene generally enhances plant growth and the creation of conductive tissues and the suppression of the *AtWOX14* gene results in plant dwarfism. The *AtWOX13* gene controls fruit development, the quantity of lateral root growth, and flowering time [9,29]. Intermediate clade *AtWOX11* and *AtWOX12* have identical roles in controlling the growth of adventitious roots and callus formation, while *AtWOX8* and *AtWOX9* interact to regulate the apical–basal polarity axis [35,36].

HOMEOBOX proteins are identified by their HOMEODOMAIN, which is an average of 60 amino acids organized physically with an N-terminal arm and three alpha helices that may bind DNA [37]. As shown in Figure 2, according to reports, the *WOX* genes’ HB domain comprises three helix structures with exceptionally preserved amino acids. These helix structures include Q, L, and E in helix 1, P, I, and L in helix 2, and I, N, V, F, Y, W, F, Q, N, and R in helix 3 [38].

The word “Wuschel” refers to the bristling and branched phenotype of altered plants, where ectopic meristems repeatedly form and degenerate before they mature [22]. Apart from the highly conserved homeodomain region, there is minimal sequence similarity across *WOX* proteins in the various subfamilies [39]. In several plant domains, *WOX* roles have diversified to provide unique features that regulate cell identification [40]. The *WOX* family was identified in various plants (Table 1).

## 3. Function of *WOX*s in Plant Developmental Processes

Transcription factors are an extensive group of regulatory proteins that play significant roles in different aspects such as growth and development by connecting their target genes through particular binding domains and controlling their expression [57]. Previous study has shown that the *WOX* transcription factor gene family has a function in the structuring of several early plant cell populations [58]. Components of the *WOX* protein family are crucial for the upkeep and growth of stem cells in the cambium, the lateral meristem that gives rise to all of the cellular parts of the wood [59]. Prior research has indicated that *WOX* genes are important for the advancement and progress of plants [60].

*WOX1* and *PRESSED FLOWER* (*PRS*; also called *WOX3*) facilitate development of cells at the leaf apical meristem and the growth of leaf laminae [61]. *WOX1* decreases the regulation of certain auxin downstream genes independently of auxin, and it positively controls auxin responsiveness and transport [62]. Overexpression of *WOX1* in *Arabidopsis thaliana* stunts the growth of plants and adversely impacts the establishment of the shoot apex [63]. *WOX1* genes show highly confined expression patterns inside the leaf primordium’s middle domain region during leaf development, designating this area as a hub for organizing the foliar extension [64]. The knockout of *SlLAM1* (*WOX1/STF/LAM1*) via CRISPR/Cas9-mediated genome editing in tomato plants resulted in narrow leaves and a reduced number of secondary leaflets, indicating its essential role in plant leaf development [10]. In *un-fused flower* tomato mutants, the knockout of the *UF/SlWOX1* gene leads to flower and organ lateral development defects [65]. *WOX1* works as a controller and detector that takes part in transportation of auxin polarity in leaf-limited area and detects the restriction caused by auxin transduction [66]. WUSCHEL homeobox 2 (*WOX2*) is important for regulating many aspects of plant somatic embryogenesis [67]. *WOX2* overexpression in *Arabidopsis* enhanced organogenesis and somatic embryogenesis in a portion of the first- and second-generation transgenic seedlings [5].

Previous research has shown that *WOX3* genes are essential for the lateral organs’ development of their lateral domains [6]. Through an analysis of the phenotypic and gene expression patterns of several *WOX3* mutants, recent study shows *CsWOX3* plays a negative role in cucumber fruit spine development. The knockout of the *CsWOX3* gene using the CRISPR/Cas9 system shows a significant increase in the fruit spine base while overexpression led to a decrease in the diameter compared to the wild type [68].

*WOX4* and *BRI1-EMS-SUPPRESSOR1* (*BES1*) likely work together to develop a signaling hub that controls the dynamic linkage of cambium cell differentiation and proliferation in response to altering phytohormonal and environmental signals [69]. *WOX4* is a key regulator of cell identity and division activity in the vascular cambium of hybrid aspen [70]. In adult *Pinus sylvestris*, *PsWOX4*’s maximum transcript level occurred during the cambial zone’s active cell proliferation phase. This tree also had the highest cambial age, 63 years, which was associated with the cambial zone’s highest number of cell layers [60]. A *WOX4* mutant shows normal xylem differentiation during secondary growth but impaired cambium cell proliferation [71]. In cotton, *GhWOX4* knockdown hindered secondary growth by reducing cambium width and division activity compared to control plants [18]. *WOX4* suppressed the expression of genes that activate gibberellin (GA), and *WOX4* overexpression was insufficient to stimulate vascular cell lignification **[9]**.

Table 2 shows the progress in studying *WOX* transcription factors in various plants. The root apical meristem (RAM) and the shoot apical meristem (SAM) of higher plants are where stem cell niches are most apparent [72]. The *WOX5* gene encodes a transcription factor, which is an essential regulator, preserving the composition and functionality of the stem cell niche in plant root tips [73]. The *WOX5* protein’s inter-cellular mobility allows it to move from the quiescent core (QC) cells to the nearby stem cells, which is crucial to control stem cells [8]. *WOX5* is essential for maintaining the stem cell niche in the root apical meristem [74]. In contrast, *WOX5* inappropriately suppresses genes associated with shoot formation, likely via inhibiting shoot growth [75]. Ectopic expression of *WOX5* in *Arabidopsis thaliana* probably represses shoot-related genes, which lead to inhibition of shoot development [73].

*WOX6/PFS2* controls ovule development and influences ovule patterning [80]. The low expression of *WOX7* resulting from low plant sugar concentration and the promoting roles of sugar and other nutrients, such as nitrogen and phosphate, prevent *WOX7*’s inhibitory effect on lateral root development under ideal growth conditions [81]. *WOX8* and *WOX9* are expressed in the zygote throughout development and in the basal offspring following zygotic division [82]. Both genes are necessary for embryo and shoot development [83]. It was reported that overexpression of the *CsWOX9* gene in cucumber regulates wart formation in cucumber fruit [84].

*WOX11* has a role in the establishment of unexpected roots (also called adventitious roots) in both *Arabidopsis* and rice [36]. *WOX11* overexpression can strengthen the capacity of falling leaves to develop roots, whereas *WOX11* suppression can result in a reduction in root formation [85]. *Arabidopsis WOX11* is essential for starting new organs by growing secondary roots from several leaves, rebuilding adventitious lateral roots from wounded primary roots, and forming calluses in tissue culture [86]. *WOX11* promotes overall plant growth and development by inhibiting nematode-induced limitation of primary root growth [87].

Plants develop more secondary roots to modify their root structure in reaction to nematode infection [88]. When the main root is broken, auxin response sites in the promoter region of *WOX11* produce a local buildup of auxin, activating the protein’s transcriptional activity [36]. The HOMEOBOX13 (*WOX13*) gene in *Arabidopsis* controls fruit patterning by suppressing the expression of the *JAG/FIL* genes in the medial domain which, in turn, permits proper replum formation [89]. *WOX13* plays a crucial role in regulating callus development and organ reconnection [79]. Research demonstrates that when tissue is injured in the leaf petiole and hypocotyl of *Arabidopsis*, *WOX13* is transcriptionally activated [79].

*WOX14* functions in the cambial zone to promote cell differentiation as compared to cell proliferation [9]. This is in opposition to prior research that hypothesized *WOX14* and *WOX4* work in coordination to promote cell proliferation [90]. Overexpression of *WOX14* in roots promotes cell differentiation, which affects stem cell maintenance [91]. In comparison to the wild type, the *WOX14* plants formed more vascular bundles in their early stems, showed a delayed blooming phenotype under long-day conditions and improved lateral root growth under short-day conditions, and decreased lignin production [9]. Figure 3 shows the role of *WOX* genes in *Arabidopsis Thaliana.*

## 4. *WOX* Targeted Genes

The *WOX* genes work in regulating other genes to activate their function. *WUS* expression initiates in the embryo at the 16-cell stage in the apical cells of domains and during post-embryonic development in the central part of the floral meristem and in the inflorescence meristem, as well as in the expressing cells of the organizing center [21]. *WUS* stimulates stem cell proliferation in floral meristem and activates the Type *2 MADS-box* gene *AGAMOUS* (*AG*). This gene determines reproductive floral organ identity and meristem development, which also works with another transcription factor, *LEAFY* (*LFY*) [92,93]. The *CLV3* gene is expressed in the central zone of the SAM and the inflorescence meristem, whereas SAM increases in size due to its function and becomes more convex due to ectopic expression of *WUS* [94].

*AGAMOUS* reduces the suppression of *WUS* transcription and plays an important role in flower meristem termination when all blooming organs have been initiated [92]. According to the flower development of the ABCDE model, *AG* is a class C gene that is responsible for the production of gynoecium and androecium. The gene SHOOTMERISTEMLESS (*STM*) gene is a key regulator of SAM development where previous research indicates that in the synchronizations of SAM, *WUS* and *STM* work side by side, and this parallel work is important for normal expression [95,96]. *WUS* expression levels increase when plants are treated with cytokinin, leading to the expansion of their expression domain, and the appearances of these plants are similar to the phenotype of plants that are mutant in the *CLAVATA* gene [97,98]. *WUS* regulates the expression of genes that are involved in auxin synthesis and response, including *TIR1*, *TAR2*/, *MP/ARF5*, and *TMO6*. *WUS* also maintains the level of auxin response in the shoot apical meristem at a stable and low level, but not at zero [99].

To maintain stem cells in RAM, *WOX5* represses the TF *CYCLING DOF FACTOR* (*CDF4*) gene, which stimulates the differentiation of columella cells by binding *TPL/TPR* factors to its promoter [8]. *OsWOX3B* and *HAIRY LEAF6* (*HL6*) promote the expression of genes involved in auxin production, signaling, and transportation, including *OsYUCCA5*, *OsPIN1* a, *OsARF4*, and *OsARF*7 [100]. *OsWOX3* interacts with *HL6* and activates it directly. Due to *TRACHERY ELEMENT DIFFERENCIATION INHIBITORY FACTORS* (*TDIF*) treatment, the expression level of *WOX4* increased in the cambium of *A. thaliana*. *TDIF* is a *CLE* peptide encoded by two *A. thaliana* genes, *CLE41* and *CLE44* [101]. Auxin increases *WOX4* expression, leading to interfascicular cambium formation and cell division inside vascular bundles [71].

*WOX7* plays an important role in root development. *WOX7* reduces the production of the cyclin gene *CYCD6*, and plants with suppression of this gene have more lateral roots than wild-type (WT) plants [81]. *CLE8* is requisite for activating *WOX8* in plants’ endosperm and suspensor. *CLE8* overexpression leads to larger seeds, while suppression of *WOX8* masks the overexpression of *CLE8* and eliminates this effect [102]. *WOX11/12* is important for callus and adventitious root formation. The mutant analysis demonstrates that *WOX11/12* activates *WOX5/7*, which plays a role in the production of adventitious roots [103]. *WOX11* also activates the expression of *LATERAL ORGAN BOUNDARIES DOMAIN* (*LBD*) TF genes, including *LBD16* and *LBD29*. These genes promote callus and adventitious root development, and a study shows that *LBD16* is directly activated by *WOX11* [7,104].

*WOX13* and *WOX14* synchronized the production of fruits, conductive tissues, and flowers. *WOX13* suppresses markers of the gynoecium’s lateral domains, including the *FILAMENTOUS FLOWER* gene, which encodes a TF from the YABBY family, and the *JAGGED* gene that encodes a zinc-finger TF [89]. *WOX14* promotes the expression of *GA3ox1* and suppresses *GA2ox1* expression, which encodes enzymes for gibberellin production and inactivation [9]. As shown in Figure 4, *WOX* gene interacts with other certain genes.

## 5. *WOX*s in Response to Abiotic Stress

### 5.1. WOXs in Response to Drought

Drought stress is a major abiotic stressor that changes plants’ cell structure, photosynthesis, and metabolism, eventually limiting plant development [105]. *WOX* transcription factors play an important role in response to drought stress. Drought treatment promotes the higher expression of *PaWOX13A* and *PaWOX13B* in roots relative to leaves, as well as in *CDR-1*, a rootstock with good drought resistance, and Gisela 5, a rootstock with a poor adaptability [106]. The expression levels of *CsWOX13*, *CsWOX14*, and *CsWOX15* were increased when tea plants faced drought and cold stress conditions [48]. *OsWOX13* increased drought tolerance in rice and also enhanced early flowering [107].

Overexpressing *OsWOX13* works with the *rab21* promoter, improves drought tolerance, and increases blooming in rice by 7–10 days [14]. Overexpression of *JcWOX5* (*Jatropha curcas*) in rice increases its vulnerability to drought stress [15]. *GhWOX4* improves drought tolerance by modulating vascular development in cotton (*Gossypium hirsutum*) [18]. Gene editing of *SlWOX4* produced by implementing the CRISPR/Cas9 system in tomato regulates the expression of genes encoding antioxidants and ABA signaling molecules, leading to improvement in drought tolerance [108]. *OsWOX11* improved drought tolerance in rice by controlling root system growth [109]. Overexpression of *MdWOX13-1* is linked to increased callus weight and improved reactive oxygen species (ROS) scavenging in response to drought stress [110].

### 5.2. WOXs in Response to Low Temperature

Low temperature can lead to cold stress, which is categorized based on the temperature. Chilling stress occurs when plants are exposed to 0–15 °C and freezing stress occurs when the temperature falls below 0 °C [111]. Plants depend on the right temperature for growth and development, and many annual flowering plants are easily affected by cold [112].

Cold stress affects plants’ photosynthesis, transpiration, respiration, germination, and flowering. Plants photosynthesize slower at low temperatures, interrupting the metabolic process. Cold stress causes yellowing of leaves and wilting, produces poor germination and stunted seedlings, and reduces tillering. Plants’ reproductive development is also affected by cold stress by delaying the reproductive stage, which results in pollen sterility. At a low temperature, the plasma membrane becomes more static and has limited fluidity, causing plants to dehydrate and damage to the membrane [111].

*AtHOS9* (*AtWOX6*) favorably enhances freezing tolerance in *Arabidopsis* [12]. After 24 h of cold treatment, pineapple (*A. comosus* L.) plants showed a highly significant expression level of the *AcoWOX13* gene [113]. In paper mulberry, the expression level of three *WOX*s was significantly raised when exposed to cold that may play an important role in cambial development [114].

### 5.3. WOXs in Response to Salinity

Most woody fruits and crop plants tend to be salt sensitive. Salinity limits the water intake and disrupts the nutrient intake, causing nutritional imbalances that cause adverse effects not only on physiological and biochemical levels but also on the molecular level of plant growth and development [115].

Overexpression of *OsWOX11* in rice improves root growth and increases the tolerance of rice to potassium deficiency [13]. *PagWOX11/12a* promotes poplar salt tolerance by activating the *PagCYP736A12* gene [16]. Overexpression of *JrWOX11* in transgenic poplar plants increases susceptibility to NaCl [17]. Bananas (*Musa acuminata* Colla.) are highly sensitive to salt, and *MaWOX3*, *MaWOX8a*, *and MaWOX11b* show maximum expression during salt treatment. This finding gives evidence that these WOX genes play a crucial role in adapting to saline conditions [1]. *SlWOX3a*, *SlWOX3b*, *SlWOX4*, *SlWOX5*, and *SlWOX13* expression patterns were significantly altered in tomato plants after 1 h of sodium chloride (NaCl) treatment, showing that they may play a role in regulating stress responses [116].

### 5.4. Expression of WOX TFs in Response to Heavy Metals

Cadmium (Cd) is a very toxic metal and highly soluble in water, being easily absorbed by plants. Through the xylem, cadmium is transported up by the roots and transferred to the aerial parts of plants, where it causes biochemical, physiological, and genetic damage [117]. In mulberry, *BpWOX2* was deeply repressed when treated with CdCl_2_, while it was highly expressed in the apical bud and leaf [19]. In *Populus*, the expression level of *PsnWOX13a* and *PsnWOX13b* was positively regulated in the early stages of CdCl_2_ treatment [118]. Cd reduced main root development and also controlled secondary root growth in *Arabidopsis* [1]. The function of *WOX11* in *Arabidopsis* helps in the adventitious lateral root production process that results from the secondary growth of the main root [119]. Transcription factor *WOX11* supports a different way that regulates the formation of spontaneous lateral roots which emerge due to tissue injury [103]. In Table 3, we summarize the role of WOX genes in response to abiotic stress.

## 6. Challenges and Future Prospects

The study of *WOX* genes is promising for future prospects, but it still faces various challenges. The *WOX* gene family shows a wide diversity across plant species, contributing to plants’ development and abiotic stress response [48]. However, limited genomic data make it difficult to implement these findings in agricultural applications, especially in non-model plants. The complex interactions of *WOX* genes with other transcription factors (e.g., *CLAVATA*) are also not well understood, which may lead to varied effects depending on the species [21].

Furthermore, the limited research on economically important plants further complicates this field. Global environmental challenges such as drought and salinity make it important to understand *WOX*-mediated stress responses more deeply. Expanding functional genomics in non-model plants by utilizing CRISPR-based gene editing and testing the functional roles of *WOX* genes in economically important crops like rice, maize, and cotton could improve crop species that are not typically model plants like *Arabidopsis* or rice. A specific focus on genes like *OsWOX13* and *GhWOX4* could reveal practical applications in crop stress tolerance and yield optimization [110]. More detailed studies on *WOX* interactions with other transcription factors, like *CLAVATA* and *BES1*, are also needed to understand developmental processes across species [120]. This may improve the manipulation of *WOX* genes to enhance crop resilience. Screening of gene-editing lines for abiotic stress responses to drought, salinity, and cold should also be prioritized. This is important for the identification of *WOX* genes that provide resilience in crops.

## 7. Conclusions

The *WOX* gene family plays an important role in regulating plant growth and development and response to environmental stresses. This review emphasizes the evolutionary distribution of *WOX* genes into different clades, each of which makes a distinctive contribution to important developmental processes such as stress adaptation, root formation, and organogenesis. The role of *WOX* genes in abiotic stress tolerance, such as to drought, salt, and cold, highlights their potential for biotechnological applications in crop resilience augmentation. However, significant problems still exist in transferring insights from model species to economically relevant crops. Future studies should focus on extending the functional genomics of *WOX* genes in non-model plants and investigating their interaction networks with other transcription factors. Using techniques like CRISPR-based gene editing to investigate *WOX* gene activities across multiple plant species would be useful in developing crops with greater resistance to environmental challenges, hence aiding sustainable agriculture.

## Figures and Tables

**Figure 1 plants-13-03108-f001:**
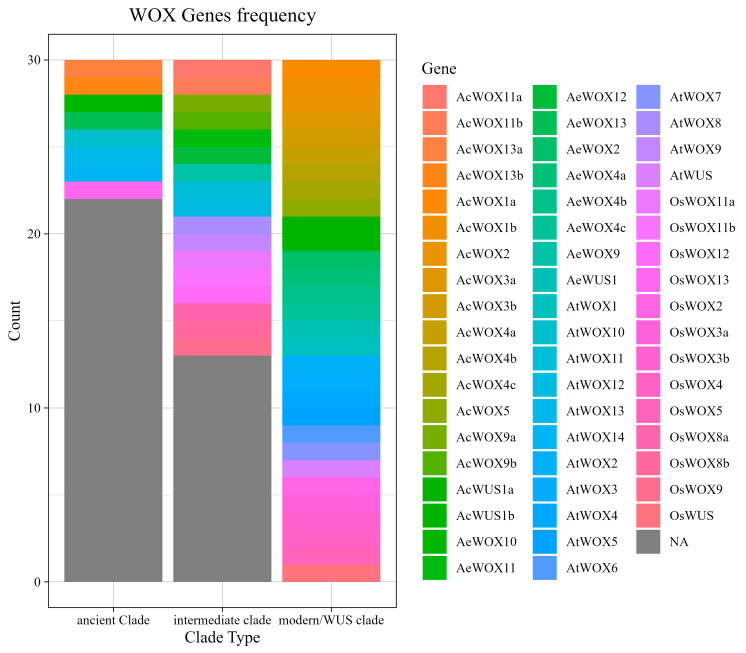
The frequency of WOX genes in three (ancient, intermediate, and modern/WUS) clades. Full-length sequences from rice (*Os*, in purple color range), *Arabidopsis* (*At*, blue color range), *A. chinensis* (*Ac*, orange color range), and *A. eriantha* (*Ae*, green color range). The modern/WUS clade is the largest clade type of the *WOX* gene family.

**Figure 2 plants-13-03108-f002:**
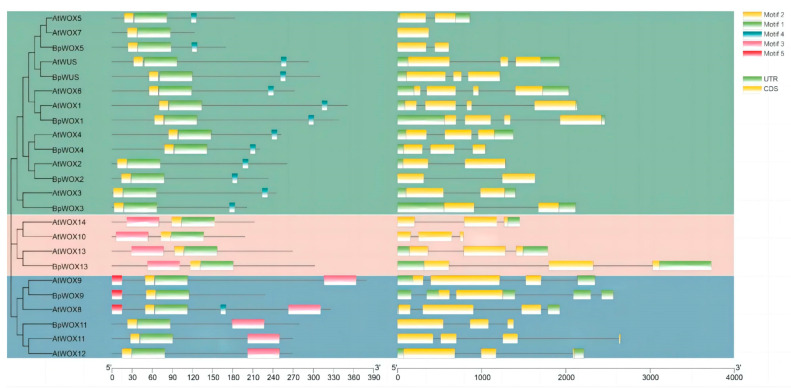
The phylogenetic analysis and gene structure comparison of the WOX gene family between *Arabidopsis thaliana* and *Betula Platyphylla*. The left-hand-side rooted tree indicates the different clades of the WOX gene family. The untranslated region (UTR) in green at the beginning and end of the gene is important for regulation and stability while the coding sequence (CDS) in yellow indicates the part of the gene that is translated into protein. Different motifs within the CDS are highlighted in different colors. Motif 1: light green, motif 2: yellow, motif 3: red, motif 4: blue, and motif 5: cyan. The horizontal scale at the bottom indicates the base pair length for each gene model, facilitating a comparison of gene size and motif lengths.

**Figure 3 plants-13-03108-f003:**
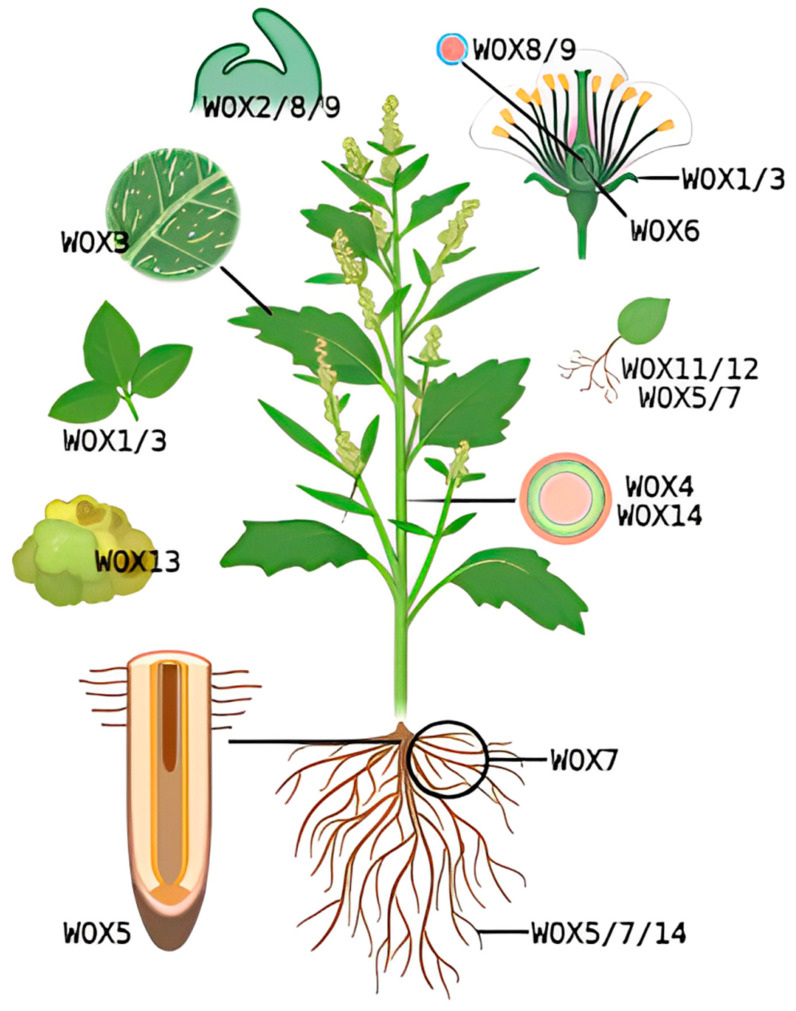
Organs and tissues of *Arabidopsis thaliana*, regarding their production and functioning, which are influenced by *WOX* TFs. This is a complete explanation of the function of *WOX* in plants’ developmental process.

**Figure 4 plants-13-03108-f004:**
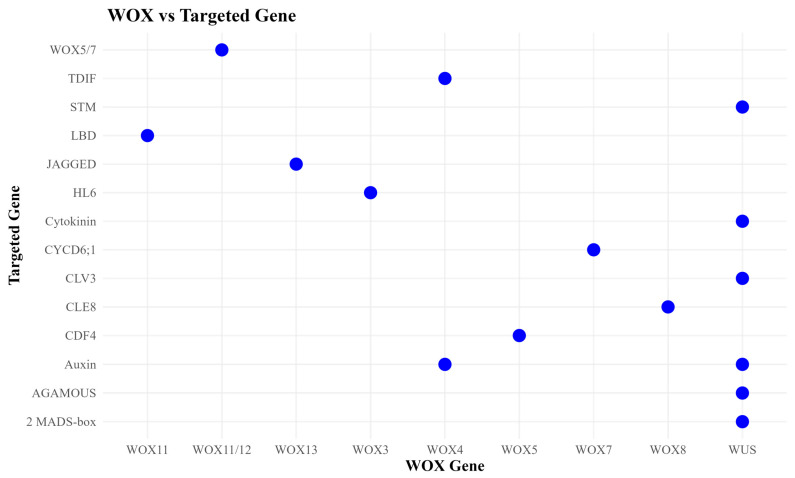
The relationship plot between the *WOX* gene and its targeted gene. The X-axis shows *WOX* gene members, and the Y-axis shows the genes that work with the *WOX* transcription factors. Genes related to auxin and cytokinin are included. The blue dot shows the target of the genes and *WOX* members.

**Table 1 plants-13-03108-t001:** WOX TFs identified in several plants.

Common Names	Latin Names	No. of Wox Genes Discovered	References
Sunflower	* Helianthus annuus *	18	[41]
Wheat	* Triticum aestivum *	14	[42]
Black cottonwood	* Populus trichocarpa *	21	[43]
European spruce	* Picea abies *	12	[44]
Maize	* Zea mays *	21	[45]
Sorghum	* Sorghum bicolor *	11	[43]
Rice	* Oryza sativa *	13	[46]
Arabidopsis	* Arabidopsis thaliana *	15	[43]
Peach	* Prunus persica *	10	[47]
Tea plant	* Camellia sinensis *	18	[48]
Coffee plant	* Cofee arabica *	7	[49]
Sweet orange	* Citrus sinensis *	11	[50]
Chinese plum	* Prunus mume *	10	[47]
Pine	* Pinus pinaster *	14	[51]
Chinese red pine	* Pinus tabulaeformis *	12	[52]
Maiden-hair tree	* Ginkgo biloba *	7	[53]
Cotton	* Gossypium arboreum *	21	[54]
Blueberry Walnut	*Vaccinium angustifolium* *Juglans regia L.*	12 12	[55] [56]

**Table 2 plants-13-03108-t002:** Function of WOX TFs in plant growth.

Common Name	Latin Name	Gene	Role of Gene	References
Arabidopsis	* Arabidopsis thaliana *	*OEWOX1*	Stunted plant growth and establishment of shoot apex	[63]
European spruce	* Picea abies *	*PaWOX2*	Protoderm development and suspensor expansion	[76]
Conifer	* Conifero phyta *	*PrWOX2*	Works as molecular marker for identifying embryogenic culture	[77]
Cotton	* Gossypium hirsutum *	*GhWOX4*	Knockdown of gene hindered secondary growth by reducing cambium width and division activity	[18]
Karelian birch	* Betula pendula *	*BpWOX4*	Wood is characterized by a more intensive growth	[78]
Moss	* Physcomitrella patens *	*PpWOX13*	required for the initiation of cell growth specifically during stem cell formation	[79]
Populus tree	* Populus alba *	*PttWOX4*	Controls cell division activity in the vascular cambium, and hence growth in stem girth	[70]
Scotch pine	* Pinus sylvestris *	*PsWOX4*	High expression that was observed during the period of active formation of early tracheid	[59]
Moss	* Physcomitrella patens *	*PpWOX13L*	Involved in cellular reprogramming at wound sites	[79]
Poplar	* Populus trichocarpa *	*PtoWOX4s* and *PtoWUSs*	Involved in vascular development	[34]
Cucumber	* Cucmis sativus *	*CsWOX1-OE*	Affects vein patterning and produces “butterfly-shaped” leaves	[66]
Conifer	* Conifero phyta *	*PpWOX2*	Affects embryogenesis-related traits	[5]

**Table 3 plants-13-03108-t003:** Role of WOX genes in response to abiotic stress.

Common Name	Latin Name	Gene	Role of Gene	References
Sweet cherry	*Prunus avium L.*	*PaWOX13A* and *PaWOX13B*	A rootstock with good drought resistance	[106]
Tea	* Camellia sinensis *	*CsWOX13*, *14*, and *15*	Regulates plant resistance in drought condition	[48]
Rice	* Oryza sativa *	*OsWOX13* *JcWOX5* *OsWOX11*	-Increases drought tolerance and enhances early flowering - Increases vulnerability to drought stress - Improves drought tolerance in rice by controlling root system growth, increases the tolerance to potassium deficiency due to salinity	[15,113] [11,109]
Cotton	* Gossypium hirsutum *	*GhWOX4*	Improves drought tolerance by modulating vascular development	[18]
Rosaceae	* Malus domestica *	*MdWOX13-1*	Increases callus weight and improves reactive oxygen species (ROS) scavenging in response to drought stress	[110]
Arabidopsis	* Arabidopsis thaliana *	*AtHOS9* (*AtWOX6*)	The response of hos9-1 mutant AtWOX6 positively regulated freezing tolerance	[78]
Pineapple	*Ananas comosus L.*	*AcoWOX13*	Shows highly significant expression after 24 h treatment in response to cold	[70]
Poplar 84K	*Populus alba x P. tremula var. glandulosa *	*PagWOX11/12a* *JrWOX11*	- Activates *PagCYP736A12* gene to promote salt tolerance - Increases susceptibility to NaCl	[11,17]
Banana	*Musa acuminata*	*MaWOX3*, *MaWOX8a*, and *MaWOX11b*	Regulates salt tolerance	[1]
Tomato	*Solanum lycopersicum*	*SlWOX3a*, *3b*, *4*, *5*, and *13* *SlWOX4*	- Regulates stress tolerance during NaCl treatment - Reduces water loss rate and enhances stomatal closure to improve drought tolerance	[114,120]
Hybrid Populus	* Populus x xiaohei *	*PsnWOX13a* and *PsnWOX13b*	Regulates stress tolerance under CdCl_2_ treatment	[118]

## Data Availability

Data are contained within the review article.

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
