# Peer review of "Plant Growth Regulators: An Overview of WOX Gene Family"

_plants, 2024, doi:10.3390/plants13213108_

Round 1

Reviewer 1 Report

Comments and Suggestions for Authors

The manuscript is good and I suggest it be published.

Author Response

The manuscript is good and I suggest it be published.

Reviewer 2 Report

Comments and Suggestions for Authors

I checked your manuscript and described comments below.

WUSCHEL-related homeobox (WOX) is a family of transcription factors important to the cell. This paper provides a very good description of the function of this gene.

I think the figures and tables in this paper are very well organized.

It would be even better if there were diagrams of the genomic structure of gene families and protein domains. I think these should be added.

I don't think this paper has major problems and grammatical problems.

Author Response

It would be even better if there were diagrams of the genomic structure of gene families and protein domains. I think these should be added.

Dear editor,

Thank you for your and reviewers’ suggestions,  It is very helpful to improve the quality of our paper. We  have revised the MS carefully. Please consider the revised MS as following:

Comment: “It would be even better if there were diagrams of the genomic structure of gene families and protein domains. I think these should be added.

Response: We would like to express our sincere gratitude for the valuable comment and suggestion. We have carefully considered your feedback and add the diagram of genomics structure of gene families and proteins domains, which is present on page 4 Of the manuscript.

Reviewer 3 Report

Comments and Suggestions for Authors

1.       The title is not adequate for the study and lacks good grammatical construct.

2.       The authors mentioned that “Studying the WOX gene diversity will help us understand the function and mechanism of the WOX gene family in plants.” In the abstract. This is not a new information.

3.       In the Introduction section, the authors elucidated the functional roles of WOX gene family in plants BUT fail to identify the bottlenecks associated with the numerous studies, particularly in different plants.

4.       The subsequent sections show the diverse roles of the WOX gene. However, these roles are available in most literatures and unfortunately the authors did not provide any new information about the subject matter.

5.       The authors would have provided the distinctions of the roles of WOX gene family in different plants, their challenges, and future roadmap on improving the relevance of the study, which involves breakthroughs in modern biotechnology.

Comments on the Quality of English Language

The grammatical construct of the manuscript is poor.

Author Response

  The title is not adequate for the study and lacks good grammatical construct.

Dear Mr.Reviwer

We would like to express our sincere gratitude for the valuable comments and suggestions. We have carefully considered your feedback and add our response according to your comments.

Comment 1: The title is not adequate for the study and lacks good grammatical construct.

Response: Thank you very much for your attention. We agree that the title needs improvement to better reflect the content of the manuscript. We have revised the title to: “Plant growth regulators: An overview of WOX gene family.”

Comment 2: “The authors mentioned, “Studying the WOX gene diversity will help us understand the function and mechanism of the WOX gene family in plants.” In the abstract. This is not a new information.”

Response: We appreciate your feedback. We have revised the sentence “Studying the WOX gene diversity will help us understand the function and mechanism of the WOX gene family in plants.” to “This review comprehensively analyzes roles of WOX gene family across various plant species, highlighting the evolutionary significance and potential biotechnological applications in stress resistance and crop improvement.”

Comment 3: “In the Introduction section, the authors elucidated the functional roles of WOX gene family in plants BUT fail to identify the bottlenecks associated with the numerous studies, particularly in different plants.”

Response: Thank you so much for pointing this out. We have revised the introduction to not only discuss the functional role of the WOX gene but also highlight the limitations in current research. The added part is highlighted in yellow color in the introduction section please take a look I attached my manuscript.

Comment 4: “The subsequent sections show the diverse roles of the WOX gene. However, these roles are available in most literature and unfortunately; the authors did not provide any new information about the subject matter.”

Response: Thank you so much for pointing out this. We acknowledge that the role of genes is well-documented in the literature. This is our (Review article) in this article we try to sum up all literature in one paper which will help to reader to review this gene effectively. We also mentioned some challenges about this gene, which are highlighted in yellow color in the manuscript.

Comment 5: “The authors would have provided the distinctions of the roles of WOX gene family in different plants, their challenges, and future roadmap on improving the relevance of the study, which involves breakthroughs in modern biotechnology”.

Response: Thank you for this suggestion. We added a new ( Table 3 ) which is present on page 12 of the manuscript in the WOX response to abiotic stress section. We also added the CRISPR gene editing technique for the WOX gene, which is very limited and highlighted in yellow text. We also revised our Conclusion section.

Comment 6: “The grammatical construct of the manuscript is poor.”

Response: We have thoroughly revised the manuscript for grammatical accuracy and improved the overall structure of our manuscript. The English version of this MS has been modified professionally.

We are looking forward to your further feedback. Thank you once again for your constructive comments, which have greatly improved the manuscript.

Round 2

Reviewer 3 Report

Comments and Suggestions for Authors

The authors should discuss the critical challenges of the current topic reviewed and prospects in a separate section. The prospects should have a detailed roadmap that addresses the individual challenges. For example; section 6. Challenges and prospects, followed by section 7. Conclusion.

Comments on the Quality of English Language

Minor English correction is required.

Author Response

The authors should discuss the critical challenges of the current topic reviewed and prospects in a separate section. The prospects should have a detailed roadmap that addresses the individual challenges. For example; section 6. Challenges and prospects, followed by section 7. Conclusion.

Response:

We would like to express our sincere gratitude for the valuable comments and suggestions. We have carefully considered your feedback and added our response according to your comments. We added a new section, which is section 6 on page 12 of our manuscript Challenges and Future Prospects. We attached our manuscript and added section highlighted in yellow color.

Comment 2

Minor English correction is required

Response:

We have thoroughly revised the manuscript for grammatical accuracy and improved the overall structure of our manuscript. The English version of this MS has been modified professionally.

We are looking forward to your further feedback. Thank you once again for your constructive comments, which have greatly improved the manuscript.

Round 3

Reviewer 3 Report

Comments and Suggestions for Authors

Remove the full meaning of WOX gene from the conclusion.

Author Response

Remove the full meaning of WOX gene from the conclusion.

We would like to express our sincere gratitude for your valuable comment and suggestion. We have carefully considered your feedback and removed the full meaning of the the WOX gene from our conclusion. Uploaded an updated version of our manuscript
